# The Effect of a Phytobiotic, Probiotic, and Their Combination as Feed Additives on Growth Performance of Weaned Holstein Male Dairy Calves

**DOI:** 10.3390/ani15081166

**Published:** 2025-04-18

**Authors:** Ting Liu, David P. Casper, Jiang Hu

**Affiliations:** 1College of Animal Science and Technology, Gansu Agricultural University, No. 1 Yingmen Village Anning, Lanzhou 730070, China; huj@gsau.edu.cn; 2Casper’s Calf Ranch, 4890 West Lily Creek Road, Freeport, IL 61032, USA; david.casper10@jcwifi.com; 3Department of Animal Sciences, North Carolina Agriculture & Technical State University, Greensboro, NC 27411, USA

**Keywords:** calves, essential oils, growth performance, phytobiotic extract, probiotic

## Abstract

A phytobiotic extract shown to enhance dry matter intake and gut health and a direct-fed microbial, when fed alone, were shown to improve gut health and growth performance. However, feeding in combination may be synergistic to post-weaning growth. Seventy-seven weaned Holstein male calves were randomly assigned to four treatments using a randomized complete block design. Calves fed the probiotic demonstrated greater body weights than calves fed control and phytobiotic with calves fed the combination being intermediate and similar. Feeding a probiotic improved body weight gains but the phytobiotic alone or in combination was not beneficial.

## 1. Introduction

The period of pre-weaning through post-weaning when transitioning calves from milk to dry feed can be very challenging [1]. Disease challenges or outbreaks can easily occur during this period [2]. While antibiotics have been commonly used during this period, the concern of antibiotic resistance has resulted in the reduction or elimination of sub-therapeutic antibiotic feeding to prevent calf disease challenges [3]. Natural plant extracts, as ingredients, are being formulated and blended into proprietary products for use as feed additives. These feed additives are actively being studied as stand-alone additives or in various feed combinations with other feed additives as potential antibiotic alternatives. These feed additives have potential to maintain or improve growth performance. Raising healthy calves affects dairy or beef operations’ long-term profitability [4].

Phytobiotic extracts (PEs) can be valuable alternative feed additives for enhancing calf growth and health [5,6,7,8]. Various PE and essential oils (EOs) have been proposed as feed additives as having antimicrobial, antiviral, and antifungal characteristics [8,9]. Reddy et al. reported nutrient digestibility, glucose, and performance benefits, but there are challenges to PEs’ appropriate use [4]. The addition of feed additives to calf starters (CSs) is proposed to benefit rumen development and accelerate growth rates [5,10,11]. In contrast, recent studies have demonstrated that specific feed additives fed alone may benefit growth performance, but when fed in combination there was no additional benefit to enhancing growth performance [12,13,14]. In fact, for example, our group has reported antagonisms between specific feed additives [13,14].

ExCell (EX) is a natural *Lactobacillus acidophilus* fermentation probiotic product containing numerous nutritional and microbial metabolites. *Lactobacillus* fermentation products have been reported to improve calf growth performance and digestive health [10,15]. However, studies have demonstrated that the beneficial responses are largely dependent on their various natural components and supplementation with different component formulations producing divergent results [4]. Jonova et al. (2021) emphasized that combining two feed additives (inulin and *Saccharomyces cerevisiae*) can improve direct-fed microbial (DFM) implantation and survival in the gastrointestinal tract to stimulate growth by activating the metabolism of a limited number of beneficial bacteria [16].

Therefore, feeding PE and EX in combination may be synergistic for enhancing post-weaning calf growth and health compared with using stand-alone phytobiotics, prebiotics, or probiotics [10,17,18]. The study objective was to evaluate the impact of feeding post-weaning calves a PE or EX alone or in combination on calf growth performance.

## 2. Materials and Methods

Casper’s Calf Ranch, LLC is a contract research facility located in Freeport, IL, USA for conducting neonatal and grower calf studies. The facilities, treatment protocols, sample collections, experimental procedures, and statistical designs are the same or closely similar to those of previous publications [12,13,19].

### 2.1. Animals and Experiment Design

The 3 wk experiment was carried out during August 2020. The calf management and feeding programs followed the guidelines published in the “Guide for the Care and Use of Agricultural Animals in Research and Teaching” published by ADSA-ASAS-PSA (2020) in conjunction with local veterinary supervision (Lena Veterinary Clinic, Lena, IL, USA) [20]. The experiment protocol number was CCR-05-2020.

Seventy-seven post-weaned Holstein male calves, 49 days old with body weight (BW) of 73.6 ± 7.5 kg SD were blocked by BW and assigned to 4 treatments. Treatments were: (1): control, (CN), calves fed CS without PE or EX; (2): PE, calves fed CS with PE (Apex) added at 275.6 g/ton to provide 0.6 kg/d; (3); EX, calves fed CS with ExCell added at 2.2 kg/ton to provide approximately 5 g/calf per d; and (4): PEEX, calves fed CS with PE and EX added at same the rates in combination. The inclusion rates of PE and EX were based on manufacturers’ recommendations (Adisseo North America Inc., Alpharetta, GA, USA and Pacer Technologies, Murtaugh, ID, USA). The PE is a blend of garlic oil, anise oil, cinnamaldehyde, rosemary, and thyme (Apex) and the EX is a *Lactobacillus acidophilus* fermentation product (ExCell). The CS ingredient composition is given in Table 1.

Calves were housed in a calf hutch (Calf-Tel Deluxe II, Hampel, Germantown, WI, USA) measuring 220 cm × 122 cm × 138 cm that was placed on a grass pasture in an open naturally well-ventilated area and bedded with chopped wheat straw. There were approximately four rows of 20 hutches each. Each hutch had a 183 cm × 122 cm × 107 cm wire panel attached to the front with 2 bucket holders (0.65 m high) for 8 L plastic buckets. One bucket was designated for ad libitum consumption of the mini-pelleted experimental CS and the other bucket was filled with water twice daily for ad libitum intake.

### 2.2. Measurement of Feed Intake and Feed Analysis

Daily CS intake was measured individually by weighing out the CS offered and ort weights daily between 8 and 10 a.m. using a digital scale (Model ACE110, Smart Weigh Inc., Hurricane, WV, USA). On rainy days, which resulted in feed being wet, these data were eliminated and the other data for the week were used to compile a daily CS mean feed intake for the week (only one day was lost in the first week, resulting in a six-day weekly mean). Calf starter samples were collected weekly and the three weekly samples were combined into a single lot for each treatment. All samples were sent to Dairyland Laboratories (Arcadia, WI, USA) for nutrient analyses.

Calf starter samples were analyzed following AOAC (2019) [21] procedures for DM (method 930.15), CP (method 990.03), soluble protein [22], NDF (method 2002.04), ADF (method 973.18), lignin (method 973.18), NDFIN (method 2002.04 without sulfite and 976.06), ADIN (method 973.18 and 976.06), water-soluble sugar [23], starch [24], fat (method 2003.05), ash (method 942.05), Ca (method 985.01), P (method 985.01), Mg (method 985.01), K (method 985.01), S (method 923.01), Na (method 985.01), Cu (method 985.01), Fe (method 985.01), Mn (method 985.01), and Zn (method 985.01). Chloride was extracted with 0.5% nitric acid and analyzed by potentiometric titration with silver nitrate (Metrohm 848 Titrino Plus, Metrohm, Riverview, Fl). Non-fiber carbohydrates (NFCs) were calculated as follows: NFC = [100 − ((NDF − NDFIN) + CP + fat + ash)], while metabolizable energy (ME; Mcal/kg), NE_m_, and NE_g_ were calculated using the NASEM (2021) equations [25]. A subsample of each experimental CS was sent to the Agricultural Experiment Station Chemical Laboratories at the University of Missouri, Columbia for AA analysis using classical ion-exchange resolution and ninhydrin post-column derivatization as described by Bai et al. following an AOAC (2019) procedure (method 982.30E) [21,26].

### 2.3. Body and Health Measurements

Individual body weights (BWs) were measured weekly using a Wrangler Jr. digital scale (Digi-Star, LLC, Fort Atkinson, WI, USA) placed on a 1.2 m × 2.4 m sheet of 1.9 cm thick sheet of plywood towed by a John Deere 825 Gator (John Deere, Moline, IL, USA). Individual BWs were taken after the morning feeding starting at approximately 8:30 a.m. each week. At the beginning and ending of the experiment, hip height (HH) and withers height (WH) were measured using a Ketchum Teletape having a level mounted on top (Ketchum Manufacturing Inc., Brockville, ON, Canada). Body length (BL) is the straight-line distance from the front edge of the scapula (shoulder) to the ischial tuberosity (hip) of the calf and heart girth (HG) or chest girth is the circumference of the calf’s chest at its widest point, typically located behind the scapula. These were measured using a Nasco dairy calf weigh tape (Nasco, Fort Atkinson, WI, USA). Hip width (HW) is the maximum horizontal distance between the two hip bones on either side of the calf’s pelvis and was measured using a Hip-O-Meter (Elanco, Green Field, IN, USA) and readings were converted from BW to cm via a regression equation. Body frame measurements were recorded at the start (week 0) and end (week 3) of the post-weaning period. In any illness case, body temperature was measured using a rectal thermometer (Zoe + Ruth, Portland, OR) and appropriate medical treatments as prescribed by a licensed veterinarian (Drs. Brandon or Morgan Scharping, Lena, IL, USA) were administered. All health incidents and treatments were recorded during the study.

### 2.4. Weather Data

Weather data were downloaded from a local personal weather station site for Freeport, IL located at 42.301° N, 89.665° W at an elevation of 256.9 m, located approximately 5 km from the research site. The temperature–humidity index (THI) was calculated according to the equation of Vitali et al. based on the weather station data for daily minimum, average, and maximum THI values [27]. A THI value of >78 was considered to compromise calves’ welfare and calves will experience significant heat stress (HS) above a THI of 88. These ranges are for HS in calves but are based on limited literature data [28].

### 2.5. Blood Collection

Blood samples were individually collected from each calf the day before the end of the experiment via jugular venipuncture using a 10 mL Vacutainer serum separation tube (Becton, Dickson and Co., Franklin Lakes, NJ, USA) via a 20-gauge, 0.9 × 38 mm blood collection needle (Nipro Corporation, Osaka, Japan) approximately 4 h after morning feeding for serum urea nitrogen concentrations. Samples were immediately placed on ice, transferred to the laboratory, and centrifuged at 2000× *g* for 10 min at room temperature (Eppendorf 5702, Eppendorf, North America, Hauppauge, NY, USA). After centrifugation, the separated serum was pipetted into polystyrene tubes (Fujian Jiakai Plastic Products Co., Ltd., Fujian, China) and stored frozen at −20 °C until analyzed. Blood urea nitrogen (BUN) concentrations were determined in duplicate using a commercially available colorimetric assay based on the diacetyl-monoxime reaction (Urea Nitrogen Kit, Procedure 0580, Stanbio Laboratory, Boerne, TX, USA) using a microplate plate reader (BioTek Epoch, Agilent, Santa Clara, CA, USA). Absorbance was measured at 520 nm using 48-well plates (Celltreat Scientific Products, Ayer, MA, USA).

### 2.6. Statistical Analysis

All individual data were checked for normality and outliers using the UNIVARIATE procedure of SAS (version 9.4, SAS Institute Inc., Cary, NC, USA) before any statistical analyses were conducted. Box and whisker plots and the Shapiro–Wilk test were used to verify that data were normally distributed (*p* > 0.15). All data were then subjected to least squares ANOVA for a randomized complete block design having 4 treatments via the MIXED procedure of SAS with study week as a repeated measure ANOVA [28]. The statistical model used was:Y_ijk_ = µ + B*_i_* + T*_j_* + WK*_k_* + (T*_j_* × WK*_k_*) + COV + *e_ijk_*
where Y_ijk_ = dependent variable, µ = overall mean, B*_i_* = blocked by BW, T*_j_* = CS treatment, WK*_k_* = experimental week, (T*_i_* × WK*_k_*) = treatment by week interaction, COV = covariate (initial measurements), and *e_ijk_* = residual random error. Calf is considered random in the model as part of the residual random error. Initial BW and frame measurements were tested as covariates for their respective parameters but did not improve statistical significance (*p* > 0.15), therefore, COV was excluded from the final model. Study week (WK*_k_*) was considered a repeated measurement in time having an autoregressive covariance structure. Treatment, week, and treatment × week interactions were considered fixed effects with block as a random effect. All parameters not measured weekly were summarized utilizing the same model described above but excluding week. Significance was declared at *p* < 0.05 and trends at 0.05 ≤ *p* ≤ 0.10. Least squares means were separated by a PDIFF statement, which is a least significant difference test, when the F-test was significant (*p* < 0.05) [29]. Daily CS fed and orts measurements were compiled as weekly averages and DM intakes were calculated.

## 3. Results and Discussion

### 3.1. Feed Analyses

All experimental CSs met or exceeded the nutrient formulation specifications (i.e., >25% CP on a dry matter basis) with approximately 23% starch, while the remaining nutrient compositions were numerically similar among all experimental CSs (Table 2), which aligns with the NASEM (2021) nutrient guidelines for growing calves [25]. In addition, CS amino acid compositions among treatments were numerically similar (Table 3). These values are very similar to our previous CS nutrient and AA compositions, minimizing the variation in supplied nutrients among our calf studies [10,26].

### 3.2. Weather Data

The study time frame matched up with typical northern Illinois August weather of being warm, dry, and humid, but not excessively causing severe heat stress (Table 4). Hot weather does not maximize neonatal calf growth and reduces feed intake [19,30]. The warm weather, with only one rain event, during the experiment was consistent with typical variation and deviations for maximum, mean, and minimum daily temperatures. The calves did experience moderate heat stress with THI exceeding > 78 at times [28], however, calves did not experience severe heat stress (i.e., THI > 88) during the experiment [28]. Temperatures above 20 °C can result in respiratory water loss [31], however, calves had ad libitum access to fresh water and respiratory health and pneumonia were not present. Certain feed additives could have an impact on animals tolerating environmental stressors (i.e., weather). For example, Reddy et al. suggested that environmental stresses would create an advantage for including probiotics to improve calf performance [4].

### 3.3. Growth Performance

The interaction of treatment by week for all measured parameters was non-significant (*p* > 0.10). As expected, week was significant (*p* < 0.01) due to increasing growth with time in the experiment but will not be discussed further. A tendency (*p*< 0.08) was detected for calves fed EX to be greater (*p* < 0.05) in overall study average BW compared with calves fed CN and PE, with calves fed PEEX being similar (*p* > 0.10) and intermediate (Table 5). In contrast, the overall BW gains (final–initial) were similar (*p* > 0.10) for calves fed all treatments even though the weights of calves fed EX were numerically greater by approximately 8.7% compared with calves fed CN. Weekly study average ADF and overall study ADG (49–70 d) were similar (*p* > 0.10) for calves fed all treatments. Calves fed EX demonstrated more than a numerical 8.9% improvement in BW and ADG gains than calves fed the CN and greater than 6% improvement in BW and ADG gains than calves fed PE and PEEX. The variation in performance parameters prevented finding statistical differences, but the initial BW covariate was non-significant (*p* > 0.70). A post hoc power and sample size analysis indicated that more than adequate calf numbers were used to detect a 5% difference with >80% power. These data reject the hypothesis that feeding PE and EX in combination would be beneficial to growth performance. Olagunju et al. reported no growth response in neonatal calves fed PE, EX, and PEEX [12], however, in a follow-up study, the combination of PE and EX (PEEX) resulted in an antagonism that reduced growth performance compared with the remaining treatments [13]. The growth response by PEEX-fed calves does not support the results reported by Jonova et al. and Stefańska et al. demonstrating positive synergistic effects when two feed additives were fed in combination [7,16]. Variability in growth responses across experiments highlights the need for further research to identify conditions under which positive effects can be consistently achieved. Cangiano et al. have also reported inconsistent responses in growth performance when feeding pre- and probiotics, but the low risk with potentially positive benefits as seen in these data can be beneficial [15].

### 3.4. Dry Matter Intake and Gain to Feed

Feeding calves PE or EX alone or in combination demonstrated similar (*p* > 0.10) DMI and gain/DMI conversions compared with calves fed CN (Table 6). Given the moderate heat stress calves were experiencing, if these additives were assisting calves in tolerating heat stress, a response in DMI or feed conversion would be expected, but this was not observed. The BUN concentrations were similar for calves fed all treatments (*p* > 0.10), indicating that these feed additives were not influencing protein metabolism. Olagunje et al. reported that the combination of PE and EX resulted in an antagonism that reduced DMI that was less than in calves fed the control treatment without any additives [12,13]. No DMI reductions were observed in this study, but no improvements either. Cangiano et al. in their review stated the feeding pro- and prebiotics resulted in similar or improved feed efficiency responses [15].

### 3.5. Frame Measurements

Calves fed EX had significantly greater (*p* < 0.05) hip width (HW) and heart girth (HG) gains compared to those fed CN and PE, while PEEX-fed calves showed intermediate gains and were not significantly different (*p* > 0.10) (Table 7). The remaining frame measurement gains (i.e., HH, WH, and BL) were similar (*p* > 0.10) among calves fed all treatments. Hill et al. reported an increase in HW when feeding a PE (i.e., Apex) [5]. Fandiño et al. explained that the active components in natural additives when combined may be additive, synergistic, or antagonistic [32]. Antagonism between specific feed additives has been reported by our group [12,13,14]. The speculation is that EX enhancing frame gains could be related to improvements in rumen and intestinal gut health [15], a shift in ruminal fermentation [33], a shift in ruminal microbial community [34], or an improvement in intestinal nutrient digestion and absorption to meet the animal’s growth requirements [25].

## 4. Conclusions

Post weaned calves fed the DFM EX demonstrated greater BW, ADG, HW, and HG gains. Calves fed all treatments were similar in DMI and FC. The experimental hypothesis that the feed additives EX and PE could be synergistic is rejected. These results demonstrate that not all feed additives are synergistic and this specific combination was not beneficial to growth performance. Future research is warranted to identify those feed additives that will be synergistic or additive to further enhance calf growth performance.

## Figures and Tables

**Table 1 animals-15-01166-t001:** Ingredient composition of control (CN), phytobiotic extract (PE), probiotic (EX), and PE and EX combination (PEEX) in pelleted calf starters (CSs).

Ingredient	Treatment			
	CN	PE	EX	PEEX
	(% Inclusion in Mix)			
Wheat midds	35.34	35.31	35.12	35.09
Soybean meal	31.50	31.50	31.50	31.50
Corn, fine ground	21.50	21.50	21.50	21.50
Molasses mixer	5.50	5.50	5.50	5.50
Calcium carbonate	2.35	2.35	2.35	2.35
Corn starch	1.25	1.25	1.25	1.25
Salt	1.10	1.10	1.10	1.10
Soy oil	0.40	0.40	0.40	0.40
Bovine B premix ^1^	0.25	0.25	0.25	0.25
Clarify 0.67% larvicide (diflubenzuron) ^2^	0.20	0.20	0.20	0.20
Vitamin A, premix ^3^	0.19	0.19	0.19	0.19
Vitamin E, premix ^4^	0.16	0.16	0.16	0.16
Decquinate, 6% ^5^	0.08	0.08	0.08	0.08
Vitamin D, premix ^6^	0.07	0.07	0.07	0.07
Dairy TM premix ^7^	0.06	0.06	0.07	0.06
Cherry flavor	0.05	0.05	0.05	0.05
Selenium yeast 2000	0.01	0.01	0.01	0.01
Phytobiotic extract ^8^	------	0.03	------	0.03
Probiotic ^9^	------	------	0.22	0.22

^1^ Bovine B contains biotin 10.5 mg/kg, choline 167.6 g/kg, folic acid 50.3 mg/kg, niacin 94 g/kg, pantothenic acid 10.5 g/kg, pyridoxine 4.4 mg/kg, riboflavin 4.4 g/kg, thiamine 1.1 g/kg, and vitamin B12 19.8 mg/kg. ^2^ Centrial Life Sciences, Schaumberg, IL, USA. ^3^ Vitamin A premix contains 11,050,072 IU/kg. ^4^ Vitamin E premix contains 44,750 IU/kg. ^5^ Zoetis Inc., Florham, NJ, USA. ^6^ Vitamin D premix contains 8,375,055 IU/kg. ^7^ Dairy trace mineral premix contains cobalt 1350 mg/kg, copper 23,500 mg/kg, iodine 2000 mg/kg, manganese 100,000 mg/kg, selenium 510 mg/kg, and zinc 125,000 mg/kg. ^8^ Apex, Adisseo North American, Alpharetta, GA, USA. ^9^ ExCell, Pacer Technologies, Murtaugh, ID, USA.

**Table 2 animals-15-01166-t002:** Nutrient composition ^1^ of control (CN), phytobiotic extract (PE), probiotic (EX), and PE and EX combination (PEEX) mini-pelleted calf starters (CSs).

Nutrient	Calf Starter			
	Control	PE	EX	PEEX
N ^2^	1	1	1	1
DM ^3^, %	88.4	87.2	86.6	86.2
CP ^4^, %	25.0	25.2	25.6	25.4
SP ^5^, % of CP	37.1	33.4	32.0	31.0
ADF, %	7.04	7.39	6.98	7.41
NDF, %	18.2	20.5	20.5	20.7
ADF-ICP	1.40	1.75	1.61	1.31
NDF-ICP	4.88	3.94	4.30	3.94
Lignin, %	7.59	7.19	5.59	5.36
NFC ^6^, %	45.7	43.6	43.5	43.5
Starch, %	22.2	23.9	23.0	23.5
Crude fat, %	3.91	3.64	3.74	3.98
ME ^7^, Mcal/kg	2.82	2.78	2.81	2.83
Nem ^8^, Mcal/kg	2.01	1.98	2.00	2.02
Neg ^9^, Mcal/kg	1.35	1.33	1.35	1.37
Ash, %	9.36	9.20	9.39	8.79
Ca, %	1.86	1.72	1.80	1.55
P, %	0.81	0.75	0.78	0.77
Mg, %	0.29	0.28	0.30	0.27
K, %	1.49	1.48	1.60	1.57
S, %	0.24	0.24	0.25	0.24
Na, %	0.38	0.30	0.43	0.34
Cl, %	0.70	0.68	0.77	0.68
Boron, ppm	14	14	15	15
Al, ppm	160	254	175	146
Fe, ppm	175	213	174	150
Mn, ppm	172	1693	145	127
Zn, ppm	185	188	193	150
Cu, ppm	27	27	28	24

^1^ Analyses conducted by Dairyland Laboratory. ^2^ N = number of samples. ^3^ Dry matter. ^4^ Crude protein. ^5^ Soluble protein. ^6^ Non-fiber carbohydrate. ^7^ Metabolizable energy. ^8^ Net energy maintenance. ^9^ Net energy gain.

**Table 3 animals-15-01166-t003:** Amino acid composition ^1^ of control (CN), phytobiotic extract (PE), probiotic (EX), and PE and EX combination (PEEX) mini-pelleted calf starters (CSs).

Amino Acid	Treatment			
	CN	PE	EX	PEEX
N ^2^	1	1	1	1
DM, %	88.6	88.3	87.4	87.4
	----------------------------- (% of DM) -----------------------------			
Arg	1.71	1.72	1.77	1.69
His	0.68	0.67	0.69	0.67
Ile	1.09	1.08	1.12	1.08
Leu	1.93	1.92	1.94	1.81
Lys	1.46	1.43	1.48	1.43
Met	0.36	0.37	0.36	0.35
Phe	1.26	1.25	1.28	1.24
Thr	0.94	0.94	.096	0.92
Trp	0.30	0.30	0.33	0.33
Val	1.25	1.20	1.25	1.21
Total EAA	11.0	10.9	11.2	10.8
Ala	1.19	1.17	1.20	1.15
Asp	2.54	2.56	2.62	2.50
Cys	0.42	0.44	0.43	0.42
Gly	1.14	1.12	1.15	1.09
Glu	4.58	1.62	1.68	4.50
Pro	1.38	1.41	1.40	1.38
Ser	1.06	1.04	1.04	1.00
Tyr	0.76	0.80	0.78	0.75
Total NEAA	13.1	13.2	13.3	12.8
Total AA	24.3	24.3	24.8	23.9
CP, %	26.1	25.3	26.2	26.6

^1^ Analyses conducted by University of Missouri. ^2^ Number of samples.

**Table 4 animals-15-01166-t004:** Weather data for the 3-week experimental period.

	Temperature, °C			Humidity, %			Wind Speed, km/h			THI, °C ^1^			Rain, cm	
Week	Max	Mean	Min	Max	Mean	Min	Max	Mean	Min	Max	Mean	Min	Mean	Total
1	29.4	23.1	16.7	89.7	68.7	46.1	25.3	9.8	1.1	83.4	70.9	61.1	0.19	1.32
2	29.4	22.9	16.5	87.3	61.7	39.9	20.9	9.2	0.0	83.1	70.0	60.5	0.00	0.00
3	31.7	25.0	18.3	87.7	64.1	41.0	20.2	9.1	0.0	87.0	73.2	62.7	0.00	0.00
Mean	30.2	23.7	17.2	88.2	64.8	42.3	22.1	4.3	0.0	84.5	77.2	61.5	0.06	1.32
SD	2.11	2.13	2.90	2.66	6.07	9.11	8.44	3.18	1.76	3.51	3.22	3.66	0.28	0.28

^1^ Temperature humidity index.

**Table 5 animals-15-01166-t005:** Body weight (BW) and average daily gain (ADG) for calves fed control (CN), phytobiotic extract (PE), probiotic (EX), and PE and EX combination (PEEX) mini-pelleted calf starters (CSs).

Measurement	Treatment					*p*< ^1^
	CN	PEX	EX	PEEX	SEM	Treatment
N	19	18	20	20	-	
BW, kg						
Week 0, initial	72.3	71.0	74.7	74.8	1.76	0.08
Week 1	76.7	76.3	79.9	79.4		
Week 2	85.6	84.2	87.7	87.0		
Week 3, final	92.7	91.7	96.8	95.6		
Study average	81.3 ^b^	80.8 ^b^	84.8 ^a^	84.2 ^ab^	1.67	0.08
Study BW gain, kg	20.3	20.7	22.1	20.8	1.14	0.61
ADG, g/d						
Week 1	628.7	754.3	737.8	658.6	106.0	0.60
Week 2	967.0	1131.4	1118.7	1078.6		
Week 3	1302.4	1067.1	1299.7	1230.0		
Study average	966.0	984.3	1052.1	989.1	63.1	0.65
Overall ADG, 49–70 d	965.5	984.3	1052.8	989.1	36.3	0.60

^1^ of F-test for treatment. ^a^,^b^ Means within the same row with unlike superscripts differ, *p* < 0.05.

**Table 6 animals-15-01166-t006:** Dry matter intake (DMI), feed conversions, and blood urea nitrogen (BUN) concentrations for calves fed control (CN), phytobiotic extract (PE), probiotic (EX), and PE and EX combination (PEEX) mini-pelleted calf starters (CSs).

Measurement	Treatment					*p*< ^1^
	CN	PE	Treatment	PEEX	SEM	Treatment
N	19	18	20	20	-	
DMI, kg/d						
Week 1	1.66	1.61	1.75	1.65	0.15	0.66
Week 2	2.18	2.22	2.36	2.17	0.15	
Week 3	2.72	2.71	2.82	2.59	0.15	
Study average	2.19	2.18	2.31	2.14	0.11	
Gain/DMI, kg/kg						
Week 1	0.40	0.47	0.40	0.39	0.05	0.24
Week 2	0.45	0.51	0.48	0.49	0.05	
Week 3	0.50	0.39	0.46	0.48	0.05	
Study average	0.45	0.46	0.45	0.45	0.02	
BUN, mg/dL	14.7	14.6	14.9	13.4	0.81	0.17

^1^ Probability of F-test for treatment.

**Table 7 animals-15-01166-t007:** Body frame measurements for calves fed control (CN), phytobiotic extract (PE), probiotic (EX), and PE and EX combination (PEEX) mini-pelleted calf starters (CSs).

Measurement	Treatment					*p*< ^1^
	CN	PE	Treatment	PEEX	SEM	Treatment
N	19	18	20	20	-----	
Hip height						
Initial, cm	90.5	90.1	90.8	90.4	0.79	0.87
Final, cm	94.8	93.8	95.2	94.7	0.71	0.27
Gain, cm	4.14	3.71	4.48	4.32	0.53	0.63
Hip width						
Initial, cm	24.4	24.2	24.6	24.5	0.23	0.58
Final, cm	26.0	26.1	26.9	26.6	0.38	0.13
Gain, cm	1.58 ^b^	1.85 ^b^	2.32 ^a^	2.09 ^ab^	0.21	0.05
Withers height						
Initial, cm	85.2	85.2	86.0	86.1	0.55	0.23
Final, cm	89.7	89.8	90.7	90.7	0.79	0.34
Gain, cm	4.47	4.54	4.70	4.60	0.47	0.99
Heart girth						
Initial, cm	93.9	93.8	94.8	94.5	0.70	0.48
Final, cm	101.7 ^b^	102.3 ^b^	105.1 ^a^	103.3 ^ab^	1.16	0.07
Gain, cm	7.88	8.48	10.36	8.76	0.83	0.13
Body length						
Initial, cm	62.8	62.8	62.4	63.2	0.71	0.85
Final, cm	65.9	66.8	67.1	67.5	0.77	0.39
Gain, cm	3.06	4.00	4.77	4.25	0.76	0.30

^1^ Probability of F-test for treatment. ^a^,^b^ Means within the same row with unlike superscripts differ, *p* < 0.05.

## Data Availability

Please contact the corresponding author for data access.

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
