# Peer review of "The Effect of a Phytobiotic, Probiotic, and Their Combination as Feed Additives on Growth Performance of Weaned Holstein Male Dairy Calves"

_animals, 2025, doi:10.3390/ani15081166_

Round 1
Reviewer 1 Report
Comments and Suggestions for Authors
Review report,
Thank you for submitting your manuscript to Animals Journal. I have carefully reviewed your work and appreciate the effort put into the study. However, several areas require revision to improve the clarity, accuracy, and readability of your paper. Below, I have outlined specific points that need attention.
In the Simple Summary section: Replace ,,compared with”…to ,,than”.
The abstract section contains some awkward phrasing and redundant details. Some sections need better flow and readability. For example, you write ADG was ,,similar” (P < 0.10) which is ambiguous, because P < 0.10 usually suggests a trend, not complete similarity.
Please reformulate the conclusion, it is very confusing ,,Feeding a probiotic improved BW gains of post-weaning Holstein male calves, but the PE alone or in combination with the probiotic (i.e. PEEX) was not beneficial for enhancing post-weaning growth performance”, extract the best results and the strong point of your topic. Further, the conclusion should be more explicit about the practical significance of the findings. Foe example: ,,These results suggest that feeding a probiotic (EX) improved …………, while PE ……………. alone or combined with EX, something like this.
Introduction Section
occur during this period [2]. While antibiotics have been commonly used during this period…" → "occur during this period [2]. Antibiotics have been commonly used during this period…", you repeat so many times, reformulate. Similar for ,, these feed additives….These additives.
"Phytobiotic extracts (PE; i.e. Apex) can be valuable alternative feed additives for en-hancing calf growth and health [5-8]." …….. "Phytobiotic extracts (PE; e.g., Apex) are valuable alternative feed additives for enhancing calf growth and health [5-8].
Ref. 12 and 13 is the same in the reference section. Please to reorganise the references section according to the Journal issues
[Error! Reference source not found., 13]???
All names of strains from topic writte italic.
Jonova et al. (2021) emphasized that combining 2 feed additives (inulin and Saccharomyces cerevisiae) can improve direct fed microbial (DFM) implantation and survival in the gastrointestinal tract to stimulate growth by activating the metabolism of a limited number of beneficial bacteria [Error! Reference source not found.].
Also, ExCell is sometimes written as "Excell".
"Lactobacillus fermen-tation products" → Should be "Lactobacillus fermentation products."
The hypothesis it is not clarity, needs rewording.
Materials and Methods section
Experimental procedures, please to provide the protocol number (No…..)
2.1 Section
,,Seventy-seven (77) 49 d old poste-weaned Holstein male calves” rewrite as "Seventy-seven (77) post-weaned Holstein male calves, 49 days old".
EU" appears instead of "EX" (likely a typo).
"Calves were housed in a chopped wheat straw calf hutch..."It sounds like the hutches were made of chopped wheat straw. Instead, clarify that the bedding was straw.
"Hutches were spaced 0.6 m apart in 4 rows of approximately 20 hutches per row. Slightly unclear. "Approximately four rows of 20 hutches each" is more better.
2.2. Section
On rainy days, which resulted in feed being wet, those data were eliminated…" "Those data" sounds awkward; "these data" or "data from those days" is preferable.
soluble protein [Error! Reference source not found.]??
"2 d were lost the first week for a 5-d weekly mean" → "Two days were lost in the first week, resulting in a five-day weekly mean."
"3 weekly samples were composited into 1 lot" → "Three weekly samples were combined into a single lot."
2.3. Section
Some abbreviations are introduced without context (e.g., BW is fine, but BL, HG, HW need a clearer introduction).
Body frame measurements were taken at the same time as BW, but only 0 and 3 wk post-weaning. Better phrased as "Body frame measurements were recorded at the start (week 0) and end (week 3) of the post-weaning period."
2.4. Section
"at an elevation of 256.9 m being approximately 5 km from the research location."
→ The phrase "being approximately" is unnecessary and disrupts readability. A clearer structure is:"at an elevation of 256.9 m, located approximately 5 km from the research site.
2.5. Section
10-mL Vacutainer serum separation tube"…. Use "10 mL" instead of "10-mL" for consistency.
"20 gauge 0.9 by 38 mm blood collection needle"→ Should be "20-gauge, 0.9 × 38 mm blood collection needle" (standardized notation).
"Once serum was separated, the serum was pipetted…"
"After centrifugation, the separated serum was pipetted…" (avoids repetition)
"Color absorbance was read at 520 nm using 48 well plates…"
"Absorbance was measured at 520 nm using 48-well plates…" (concise and correct phrasing)
2.6. Section
"Box-and-whisker plots and the Shapiro-Wilk test were used…" (plural agreement correction)
"Initial BW and frame measurements was tested as a covariate…"
"Initial BW and frame measurements were tested as covariates…" (plural correction)
3. Section
3.1. …..
All experimental CS met or exceeded nutrient formulation specifications (i.e. > 25% CP, DM basis)...→ "All experimental calf starters (CS) met or exceeded the nutrient formulation specifications (i.e., >25% CP on a dry matter basis)..." (Clarifies CS abbreviation early and improves flow.)
,, with approximately 23% starch and the remaining nutrients being similar (P > 0.10) across all experimental CS (Table 2)"→ "with approximately 23% starch, while the remaining nutrient composition was similar (P > 0.10) among all experimental CS (Table 2)."
"which would meet the NASEM (2021) growing calf nutrient guidelines [25]" "which aligns with the NASEM (2021) nutrient guidelines for growing calves [25]." (More precise and concise.)
CS AA composition among treatments were similar (Table 3). Clarihy AA abbreviation
3.2.
"Temperatures above 20°C can result in respiratory water loss [Error! Reference source not found.]"→ The reference error must be corrected or removed.
3.3
"Calves fed EX were greater (P < 0.05) in overall study average BW…
"Calves fed EX had a significantly greater average BW (P < 0.05) compared to those fed CN and PE, while calves fed PEEX had intermediate BW and were not significantly different (P > 0.10)." (More precise wording.)
"[Error! Reference source not found.]" appears multiple times. These references should be corrected or removed.
The inconsistencies in growth responses across experiments can be puzzling and more work is needed to accurately define the situations where positive responses can be achieved."……. "Variability in growth responses across experiments highlights the need for further research to identify conditions under which positive effects can be consistently achieved."
In Table 5. The lowercase write as superscripts
3.5.
Calves fed EX demonstrated greater (P < 0.05) HW and HG gains compared with calves fed CN and PE with calves fed PEEX being similar (P > 0.10) and intermediate (Table 7)."……….. "Calves fed EX had significantly greater (P < 0.05) hip width (HW) and heart girth (HG) gains compared to those fed CN and PE, while PEEX-fed calves showed intermediate gains and were not significantly different (P > 0.10) (Table 7)."
"But why feeding EX would enhance frame gains is speculated to be related to improvements in rumen and intestinal gut health [15], shift in ruminal fermentation [33], shift in ruminal microbial community [34], or an improvement in intestinal nutrient digestion and absorption to meet the animal’s growth requirements [25]."
→ The phrase "But why feeding EX would enhance frame gains is speculated" is awkward. Revised!
Improve the Conclusion section
Author Response
Reviewer 1
Thank you for submitting your manuscript to Animals Journal. I have carefully reviewed your work and appreciate the effort put into the study. However, several areas require revision to improve the clarity, accuracy, and readability of your paper. Below, I have outlined specific points that need attention.
In the Simple Summary section: Replace ,,compared with”…to ,,than”.
Re:Thank you very much for your comments. We have already made the corrections.
The abstract section contains some awkward phrasing and redundant details. Some sections need better flow and readability. For example, you write ADG was ,,similar” (P < 0.10) which is ambiguous, because P < 0.10 usually suggests a trend, not complete similarity.
Re:Thank you. We have revised it according to your suggestions.
Please reformulate the conclusion, it is very confusing ,,Feeding a probiotic improved BW gains of post-weaning Holstein male calves, but the PE alone or in combination with the probiotic (i.e. PEEX) was not beneficial for enhancing post-weaning growth performance”, extract the best results and the strong point of your topic. Further, the conclusion should be more explicit about the practical significance of the findings. Foe example: ,,These results suggest that feeding a probiotic (EX) improved …………, while PE ……………. alone or combined with EX, something like this.
Re:I agree with your opinion and have already made the revisions in the manuscript.
Introduction Section
occur during this period [2]. While antibiotics have been commonly used during this period…" → "occur during this period [2]. Antibiotics have been commonly used during this period…", you repeat so many times, reformulate. Similar for ,, these feed additives….These additives.
"Phytobiotic extracts (PE; i.e. Apex) can be valuable alternative feed additives for en-hancing calf growth and health [5-8]." …….. "Phytobiotic extracts (PE; e.g., Apex) are valuable alternative feed additives for enhancing calf growth and health [5-8].
Re:Okay, we've removed the duplication.
Ref. 12 and 13 is the same in the reference section. Please to reorganise the references section according to the Journal issues
Re:Thank you for your comments, the reference has been confirmed.
[Error! Reference source not found., 13]???
Re:The error has since been removed from the manuscript.
All names of strains from topic writte italic.
Re:Already modified.
Jonova et al. (2021) emphasized that combining 2 feed additives (inulin and Saccharomyces cerevisiae) can improve direct fed microbial (DFM) implantation and survival in the gastrointestinal tract to stimulate growth by activating the metabolism of a limited number of beneficial bacteria [Error! Reference source not found.].
Re:Already modified.
Also, ExCell is sometimes written as "Excell".
Re:Thank you. It has been modified.
"Lactobacillus fermen-tation products" → Should be "Lactobacillus fermentation products."
Re:Thank you. It has been modified.
The hypothesis it is not clarity, needs rewording.
Re:Ok, we have modified it.
Materials and Methods section
Experimental procedures, please to provide the protocol number (No…..)
Re:Thank you for your comments, we have added the relevant numbers to the manuscripts.
2.1 Section
,,Seventy-seven (77) 49 d old poste-weaned Holstein male calves” rewrite as "Seventy-seven (77) post-weaned Holstein male calves, 49 days old".
Re:Thank you for your comments, we have changed it.
EU" appears instead of "EX" (likely a typo).
Re:Thank you. We have revised it according to your suggestions.
" Calves were housed in a chopped wheat straw calf hutch..."It sounds like the hutches were made of chopped wheat straw. Instead, clarify that the bedding was straw.
Re:No, the cow's cage was covered with straw to keep it dry and warm.
"Hutches were spaced 0.6 m apart in 4 rows of approximately 20 hutches per row. Slightly unclear. "Approximately four rows of 20 hutches each" is more better.
Re:Thank you, we have modified according to your comments.
2.2. Section
On rainy days, which resulted in feed being wet, those data were eliminated…" "Those data" sounds awkward; "these data" or "data from those days" is preferable.
Re:Already modified
soluble protein [Error! Reference source not found.]??
Re:We are very sorry for this problem, but we have solved it in the article
"2 d were lost the first week for a 5-d weekly mean" → "Two days were lost in the first week, resulting in a five-day weekly mean."
Re:Thank you for your comments, we have changed it.
"3 weekly samples were composited into 1 lot" → "Three weekly samples were combined into a single lot."
Re:Thank you for your comments, we have changed it.
2.3. Section
Some abbreviations are introduced without context (e.g., BW is fine, but BL, HG, HW need a clearer introduction).
Re:Well, thank you for reminding us that we have included an explanation of them in the manuscript.
Body frame measurements were taken at the same time as BW, but only 0 and 3 wk post-weaning. Better phrased as "Body frame measurements were recorded at the start (week 0) and end (week 3) of the post-weaning period."
Re:Thank you for your comments, we have changed it.
2.4. Section
"at an elevation of 256.9 m being approximately 5 km from the research location."
→ The phrase "being approximately" is unnecessary and disrupts readability. A clearer structure is:"at an elevation of 256.9 m, located approximately 5 km from the research site.
Re:Thank you for your comments, we have changed it.
2.5. Section
10-mL Vacutainer serum separation tube"…. Use "10 mL" instead of "10-mL" for consistency.
Re:Thank you for your comments, we have changed it.
"20 gauge 0.9 by 38 mm blood collection needle"→ Should be "20-gauge, 0.9 × 38 mm blood collection needle" (standardized notation).
Re:Thank you for your comments, we have changed it.
"Once serum was separated, the serum was pipetted…"
Re:Thank you for your comments, we have changed it.
"After centrifugation, the separated serum was pipetted…" (avoids repetition)
Re:Thank you for your comments, we have changed it.
"Color absorbance was read at 520 nm using 48 well plates…"
"Absorbance was measured at 520 nm using 48-well plates…" (concise and correct phrasing)
Re:Thank you for your comments, we have changed it.
2.6. Section
"Box-and-whisker plots and the Shapiro-Wilk test were used…" (plural agreement correction)
Re:Thank you for your comments, we have changed it.
"Initial BW and frame measurements was tested as a covariate…"
"Initial BW and frame measurements were tested as covariates…" (plural correction)
Re:Thank you for your comments, we have changed it.
- Section
3.1. …..
All experimental CS met or exceeded nutrient formulation specifications (i.e. > 25% CP, DM basis)...→ "All experimental calf starters (CS) met or exceeded the nutrient formulation specifications (i.e., >25% CP on a dry matter basis)..." (Clarifies CS abbreviation early and improves flow.)
Re:Thank you for your comments, we have changed it.
,, with approximately 23% starch and the remaining nutrients being similar (P > 0.10) across all experimental CS (Table 2)"→ "with approximately 23% starch, while the remaining nutrient composition was similar (P > 0.10) among all experimental CS (Table 2)."
Re:Thank you for your comments, we have changed it.
"which would meet the NASEM (2021) growing calf nutrient guidelines [25]" "which aligns with the NASEM (2021) nutrient guidelines for growing calves [25]." (More precise and concise.)
Re:Thank you for your comments, we have changed it.
CS AA composition among treatments were similar (Table 3). Clarihy AA abbreviation
Re:Okay, we changed the name of it
3.2.
"Temperatures above 20°C can result in respiratory water loss [Error! Reference source not found.]"→ The reference error must be corrected or removed.
Re:Well, it's been removed from the manuscript.
3.3
"Calves fed EX were greater (P < 0.05) in overall study average BW…
"Calves fed EX had a significantly greater average BW (P < 0.05) compared to those fed CN and PE, while calves fed PEEX had intermediate BW and were not significantly different (P > 0.10)." (More precise wording.)
Re:Thank you for your comments, we have changed it.
"[Error! Reference source not found.]" appears multiple times. These references should be corrected or removed.
Re:OK, it's been removed from the manuscript.
The inconsistencies in growth responses across experiments can be puzzling and more work is needed to accurately define the situations where positive responses can be achieved."…….
Re:Ok, thank you for your comments.
"Variability in growth responses across experiments highlights the need for further research to identify conditions under which positive effects can be consistently achieved."
Re:Ok, thank you for your comments.
In Table 5. The lowercase write as superscripts
Re:Ok
3.5.
Calves fed EX demonstrated greater (P < 0.05) HW and HG gains compared with calves fed CN and PE with calves fed PEEX being similar (P > 0.10) and intermediate (Table 7)."……….. "Calves fed EX had significantly greater (P < 0.05) hip width (HW) and heart girth (HG) gains compared to those fed CN and PE, while PEEX-fed calves showed intermediate gains and were not significantly different (P > 0.10) (Table 7)."
Re:Thank you for your comments, we have changed it.
"But why feeding EX would enhance frame gains is speculated to be related to improvements in rumen and intestinal gut health [15], shift in ruminal fermentation [33], shift in ruminal microbial community [34], or an improvement in intestinal nutrient digestion and absorption to meet the animal’s growth requirements [25]."
→ The phrase "But why feeding EX would enhance frame gains is speculated" is awkward. Revised!
Improve the Conclusion section
Re:Ok, we have made relevant changes to the manuscript.
Reviewer 2 Report
Comments and Suggestions for Authors
The abstract has to be rechecked along with results as the conclusion says different.
Check grammar
Please see that some references are error
Cross check the results
All the relevant corrections are included in the manuscript

Can be improved
Author Response
Thank you very much for your valuable comments on this manuscript, please check the following document for changes to the comments.
review 2
1.“Calves fed the probiotic demonstrated greater body weights compared with calves fed control and phytobiotic with calves fed the combination being intermediate and similar. ” Paraphrase for better understanding.
Re:Thank you for pointing this out. We agree with this comment. Therefore, we have rewritten
this part in accordance with your suggestion.
2.“Lactobacillus acidophilus” Italics.
Re:Thank you for kindly reminding us. We have update the text as suggested.
- “Average daily gains (ADG) were similar (P < 0.10) for calves (965.5, 984.3, 1052.8 and 989.1 g/d) fed all treatments.” In conclusion your study says that ADG is more in EX group please check.
Re:Thank you for pointing this out. We have update the text as suggested.
4.“Phytobiotic extracts (PE; i.e. Apex)”Apex means? trade name?
Re:Thank you for kindly reminding us. We have update the text.
- “ The addition of feed additives to CS is proposed to benefit rumen development and accelerate growth rates as reported by Hill et al.” CS: Expand at first instance.
Re:Thank you for your reminder. We have update the text as suggested.
- “[Error! Reference source not found.,13].” Please check.
Re:We were really sorry for our careless mistakes.Thank you for your reminder.
- “Saccharomyces cerevisiae” Italics.
Re:Thank you for kindly reminding us. We have update the text as suggested.
- “The facilities, treatment protocols, sample collections, experimental procedures, and statistical designs are the same or closely simi- lar too previous publications.” to
Re:We were really sorry for our careless mistakes. We have corrected the “too” into “to”. Thanks for your correction.
- “Seventy-seven (77) 49 d old poste-weaned Holstein”.77: remove.
Re:Thanks for your careful checks. We have update the text as suggested.
- “(73.6 BW ±5 kg SD)”.with Body weight (mean +SD) (73.6+7.5).
Re:Thank you for pointing this out. We have update the text as suggested.
- “Lactobacillus acidophilus”Italics.
Re:Thank you for pointing this out.
- “Soybean meal, 48 ” 48 means.
Re:Thank you for pointing this out. We have update the text.
- “(2 d were lost the first week for a 5-d weekly mean).” Paraphrase.
Re: Thank you for kindly reminding us. We agree that this describe the process more clearly and have update the text as suggested.
- “1Analyses conducted by Dairyland Laboratory.” 1 is not indicated in the table.
Re:Thank you for your reminder. We have update the text.
- “1Analyses conducted by University of Missouri.”indicate 1 in table as well.
Re:Thank you for kindly reminding us. We have update the text.
- “however in a follow up study, the combination of PE and EX (PEEX) resulted in an antagonism that reduced growth performance compared with the remaining treatments”Check combination such as PEEX may have antagonist effect on the growth.
Re: Thanks to the reviewers for bringing this to our attention.
- “Table 5. Cont.” Remove and continue the table as single only.
Re: Thank you for kindly reminding us. We have modified the table in the text.
- “81.3b;80.8 b;84.8 a;84.2 ab;” Superscript all.
Re: Thank you for kindly reminding us. We have update the text.
- “1Probability of F test for treatment.”indicate 1 in table.
Re: Thank you for kindly reminding us. We have indicated in the table.
- “Calves fed EX demonstrated greater BW and ADG compared with calves fed CN and PE with calves fed PEEX being intermediate and similar.”Your abstract says the EX didnt have any significant ADG? please check.
Re:Thanks to the reviewers for bringing this to our attention.
- “Feeding calves the PEEX combination resulted in similar growth performance compared to feeding BE and EX individually.” “BE” it should be PE.
Re:We were really sorry for our careless mistakes. We have corrected the “too” into “to”. Thanks for your correction.
Reviewer 3 Report
Comments and Suggestions for Authors
Manuscript number: animals-3530448
Title:
The effect of a phytobiotic, probiotic, and their combination as feed additives on growth performance of weaned Holstein male dairy calves
The authors evaluate feeding post-weaning calves a PE or EX alone or in combination on calf growth performance. They reported that feeding a probiotic improved BW gains of post-weaning Holstein male calves, but PE alone or in combination with the probiotic (i.e., PEEX) was not beneficial for enhancing post-weaning growth performance. Although this work is interesting, several concerns were found throughout the manuscript and require substantial revision before further consideration.
Abstract:
- Change the abstract according to the review given!
Introduction:
- Additives are not only antibiotics, why do you state that additives can be antibiotics. The writing must be more focused and clearer in the direction it is heading. Because the definition of additive is very broad.
- There are many references that are erroneous, you should reread and check again.
- The gap in the study is not clear, the identification of why this research should be done is not clear and difficult to understand.
- From the beginning you explained about phytobiotic and probiotic why suddenly in the hypothesis you explained about prebiotic?
Material and Method
- The writing of abbreviations such as PE EX and CS should be consistent. And if it has been given an understanding of what it stands for, it should not need to be explained again. This is like a manuscript that is written PE (Apex) even though it has been explained before. There is also wrong writing such as EX becomes EU, this confuses the reader because you do not check the wrong writing again.
- I am curious, in table 1 why is PE only 0.03% and EX 0.22%? You should explain this.
- Why when it rains the data is not retrieved because it gets wet? What happens when it rains? And how many times did it rain that the data was not entered?
- Why is blood collection done but not in the result?
- There are many references that have errors, you should reread and check again.
Results and Discussion
- The explanation of feed analysis is not deep enough, there are some differences that occur in SP, but it is not discussed in more depth. There should be an explanation of the p value in table 2 so that it is easy to compare between the four.
- The explanation of weather data is not deep enough and not clear.
- In the growth performance parameter, what should be significant is p<0.05, but in the results it is said to be significant even though it has a value of 0.08 seen from the superscript a, b, c values given (Table 6; study average). If 0.08 is significant, why is table 5; body weight not given superscript a, b, c?
- There are many references that have errors, you should reread and check again.
- The explanation and discussion are just a formality, not explained in depth and only compared with the results of previous studies.
- Is there no explanation of blood profile? So why look for blood collection in materials and methods?
Conclusion
- Conclusion is unclear and does not explain the purpose correctly.
- Provide recommendations and suggestions for future research.
References:
Check format style and Journal Abbreviation! And Ensure consistency in formatting references!
Comments on the Quality of English LanguageThe English could be improved to more clearly express the research.
Author Response
Thank you very much for your valuable comments on this manuscript, please check the following document for changes to the comments.
Reviewer 3:
Title:
The effect of a phytobiotic, probiotic, and their combination as feed additives on growth performance of weaned Holstein male dairy calves
The authors evaluate feeding post-weaning calves a PE or EX alone or in combination on calf growth performance. They reported that feeding a probiotic improved BW gains of post-weaning Holstein male calves, but PE alone or in combination with the probiotic (i.e., PEEX) was not beneficial for enhancing post-weaning growth performance. Although this work is interesting, several concerns were found throughout the manuscript and require substantial revision before further consideration.
Abstract:
- Change the abstract according to the review given!
Re:
Introduction:
- Additives are not only antibiotics, why do you state that additives can be antibiotics. The writing must be more focused and clearer in the direction it is heading. Because the definition of additive is very broad.
Re:The abstract has been modified to provide clarity.
- There are many references that are erroneous, you should reread and check again.
Re:Thank you very much for your comments. We have already made the corrections.
- The gap in the study is not clear, the identification of why this research should be done is not clear and difficult to understand.
Re: Actually the reason is that feed additives are evaluated separated to a control but studies are limited when these addivites are fed in combinations.
- From the beginning you explained about phytobiotic and probiotic why suddenly in the hypothesis you explained about prebiotic?
Re:In my opinion these different nomenclatures gets a bit confusing if it’a probiotic or a prebiotic. Especially when the company’s product claims both characteristics. Then you always have the issue if the product is alive or dead.
Material and Method
- The writing of abbreviations such as PE EX and CS should be consistent. And if it has been given an understanding of what it stands for, it should not need to be explained again. This is like a manuscript that is written PE (Apex) even though it has been explained before. There is also wrong writing such as EX becomes EU, this confuses the reader because you do not check the wrong writing again.
Re:Thank you very much for your comments. We have already made the corrections.
- I am curious, in table 1 why is PE only 0.03% and EX 0.22%? You should explain this.
Re:It’s due to the different concentrations of the products technology which is given in the Materials and methods to provide the actual feeding rate. Of course realize that the products have carriers added to them. But the targeted feeding rates are given in T&M.
- Why when it rains the data is not retrieved because it gets wet? What happens when it rains? And how many times did it rain that the data was not entered?
Re:Again in section 2.2 of M&M we lost only 1 day due to rain.
- Why is blood collection done but not in the result?
Re:It was. BUN concentrations are given in Table 6 which is measurements from this test.
- There are many references that have errors, you should reread and check again.
Re:Thank you very much for your comments. We have already made the corrections. That is Dr. Casper mistakes and take responsibility, but also my compliments, great catch.
Results and Discussion
- The explanation of feed analysis is not deep enough, there are some differences that occur in SP, but it is not discussed in more depth. There should be an explanation of the p value in table 2 so that it is easy to compare between the four.
Re:I believe this to be serendipity. SBM can and does vary in SP as well as other ingredients. I believe that any speculation on our part for the SP beyond ingredient variability is a stretch. In addition, at this age, SP can’t have much of an impact other than urea synthesis and excretion being an energy cost.
- The explanation of weather data is not deep enough and not clear.
Re:I would like to challenge this statement, because what we are tyring to relay that calves did have some heat stress but it was not severed or extreme. So if these products can help/alleviate heat stress in animals, not sure we would pick it up, because heat stress was not severe. I again believe any further discussion beyond this is pure speculations. Granted the more severe the heat stress the pooper growth rates and DMI are.
- In the growth performance parameter, what should be significant is p<0.05, but in the results it is said to be significant even though it has a value of 0.08 seen from the superscript a, b, c values given (Table 6; study average). If 0.08 is significant, why is table 5; body weight not given superscript a, b, c?
Re:In the discussion we specify 0.08 as a tendency for the overall Treatment effect across the study. But when conducting means separation, we stuck with the standard 0.05. so that should explain the difference.
- There are many references that have errors, you should reread and check again.
Re:Thank you very much for your comments. We have already made the corrections.
- The explanation and discussion are just a formality, not explained in depth and only compared with the results of previous studies.
Re:In our defense, this experiment continues a series of experiments evaluating different technologies alone or in combination. If you noticed some of our previous work on essential oils and ionophores have reported antagonistic results. So again, I don’t believe the reader is wanting an in depth detail discussion when the results were not favorable and the combinations did not further enhance growth rates. Also our hypothesis was rejected. So we opted for the report of results, some discussion as to why or why not so they can make a feeding decision of a product or combination of products.
- Is there no explanation of blood profile? So why look for blood collection in materials and methods?
Re:Blood urea nitrogen data., Table 6, 1 sentence before the table stating no difference among treatments and not influencing protein metabolism. Felt that was enough.
Conclusion
- Conclusion is unclear and does not explain the purpose correctly.
Re:Thank you very much for your comments. We have already made the corrections.
- Provide recommendations and suggestions for future research.
Re: Thank you very much for your comments. We have already made the corrections.
References:
Check format style and Journal Abbreviation! And Ensure consistency in formatting references!
Re:Thank you very much for your comments. We have already made the corrections.
Round 2
Reviewer 1 Report
Comments and Suggestions for Authors
Dear Authors/Editors,
We acknowledge the authors' revisions and appreciate their thorough responses. All comments and suggestions have been adequately addressed, and the manuscript has been improved accordingly. I have no further concerns.
Best regards,
Mihaela DUMITRU
Reviewer 2 Report
Comments and Suggestions for Authors
The manuscript has been significantly improved as per the suggestions; however, the minor errors may be cross checked if any before publication.
Reviewer 3 Report
Comments and Suggestions for Authors
All my concerns were addressed, and there were no further suggestions.